# Factors associated with elder abuse and neglect in rural Uganda: A cross-sectional study of community older adults attending an outpatient clinic

**Letizia Maria Atim**[1], **Mark Mohan Kaggwa**[1,2]\*, **Mohammed A. Mamum**[3,4], **Moses Kule**[5], **Scholastic Ashaba**[1], **Samuel Maling**[1]

1 Department of Psychiatry, Faculty of Medicine, Mbarara University of Science and Technology, Mbarara, Uganda, 2 Department of Psychiatry and Behavioral Neurosciences, McMaster University, Hamilton, ON, Canada, 3 CHINTA Research Bangladesh, Savar, Dhaka, Bangladesh, 4 Department of Public Health and Informatics, Jahangirnagar University, Savar, Dhaka, Bangladesh, 5 Department of Psychiatry, Mbarara Regional Referral Hospital, Mbarara, Uganda

\* kmarkmohan@gmail.com

**Data Availability Statement:** All relevant data are within the paper and its Supporting Information files.

## Abstract

### Background

Elderlies are vulnerable to abuse, and evidence suggests that one in three elderlies experience abuse. Abuse can impact the well-being of older persons, decreasing their quality of life, leading to mental health challenges, and increasing morbidity and mortality rates. Evidence on older person/elder abuse and neglect is vital to facilitate initiatives, but there are fewer studies on elder abuse and neglect in Africa, particularly in Uganda. Thus, this study aimed to determine the prevalence of different types of abuse and neglect, and their associated factors among older persons (aged 60 years and above) attending an outpatient clinic.

### Methods

In this cross-sectional study, information on sociodemographic characteristics, functional impairment using the Barthel Index, and elder abuse severity using the Hwalek-Sengstock Elder Abuse Screening Test were collected. In addition, types of abuse were assessed using questions adapted from the US National Research Council on elder mistreatment monograph. Linear and logistic regression analyses were used to determine the factors associated with elder abuse severity and the different types of abuse, respectively.

### Results

Overall, the prevalence of elder abuse was 89.0%. Neglect was the most common type of elder abuse (86%), followed by emotional abuse (49%), financial abuse (46.8%), physical mistreatment (25%), and sexual abuse (6.8%). About 30.4% of the abused elders experienced at least two forms of abuse. Factors associated with elder abuse severity were having a secondary level of education and physical impairment. Moderate to severe functional dependence was associated with almost all forms of abuse. Individuals who reported the

**Funding:** Dr. Scholastic Ashaba declares salary support from the National Institutes of Health under Grant No. K43TW011929.

**Competing interests:** The authors have declared that no competing interests exist.

presence of a perpetrator were likely to experience neglect, emotional, and physical mistreatment. However, those who reported their perpetrators to the police had a higher likelihood of experiencing emotional abuse but were less likely to experience financial abuse. Emotional abuse was also associated with age above 80 years and attaining education (primary and secondary). Physical impairment and chronic medical conditions reduced the likelihood of experiencing neglect and financial abuse, and physical abuse, respectively.

## Conclusions

Uganda has a high prevalence of elder abuse and neglect. There is a need to design interventions for older adults at risk to prevent elder abuse from escalating further, where the present findings can be worthy of help.

## Introduction

Due to higher life expectancy, lower fertility rates, improved healthcare systems, and longevity, the aged population is increasing globally [1]. About one billion people aged 60 years or above will increase to 1.4 billion and 2.1 billion by 2030 and 2050, respectively [2]. In low-and middle-income countries (LMICs), the expected population growth rates remain higher than in high-income countries [3], and 80% of the global older adults will be in LMICs by the year 2050 [2]. However, the increasing older adult population faces physical and mental health challenges, including older persons/elder abuse [4]. For instance, in Uganda, older persons face multiple challenges, including poverty, poor health, unemployment, chronic ill-health, HIV/ AIDS, lack of social security systems, low land productivity, political instability, low agricultural returns, and functional inability [5–8]. Culturally, in Uganda, older people were highly respected; younger people and children would even kneel to greet them, and were considered a symbol of knowledge and power, but all has changed in recent years mainly due to cultural dilution [5]. The country has also been affected by high rates of domestic violence, especially towards women in rural settings, and the law has poorly handled them, thus perpetrating these unhealthy behaviors against many individuals, including older persons [9–12]. These challenges impact older person's quality of life and make them susceptible to neglect and abuse by their peers and family members [5, 13].

Elder abuse can be in different forms, such as physical, sexual, emotional/psychological, financial, abandonment, neglect, and institutional maltreatment [5, 13]. A recent systematic review of older persons from rural settings by Zhang and colleagues reported the pooled prevalence of elderly abuse at approximately 33% [14]. Among these older persons from a rural setting, 5%, 7%, 17%, and 26% experienced financial, physical, and emotional abuse, and neglect, respectively [14]. Evidence from a systematic review of past-year abuse showed abuse among 15.7% of older persons [15]; emotional abuse was dominant at 11.6%, followed by financial (6.8%), then neglect at 4.2%, and sexual abuse least experienced at 0.9% [15]. In Europe, prevalence varies from 2.2% (Ireland) and 61.1% (Croatia), whereas, in Asia, the highest prevalence was reported in China (36.2%), and the lowest was in India at 14.0%, as per a global systematic review [16]. In South Africa, men's and women's prevalence rates were 64.3% and 60.3%, respectively. Physical abuse was more common among men, while emotional, financial, and sexual abuse was more frequent among women [17].

Abused older adults present to the hospital with different signs and symptoms, including inadequately or usually unexplained locations of abrasions or bruises, lacerations, and burns

[18]. Other signs are fractures in non-alcoholics, malnutrition and dehydration, pressure ulcers, sexually transmitted infections, and vaginal and rectal bleeding without a reasonable explanation [18].

Evidence has highlighted some of the factors associated with elder abuse across the globe, including (i) perpetrator factors such as mental illness, abuser dependency, and alcohol dependence; (ii) host factors such as dementia, physical dependency, living alone, or with spouse; and (iii) environmental factors such as social isolation and negative societal perceptions towards aging [19]. In addition, psychiatric illness among older adults is an essential cause of vulnerability to abuse, especially when it is comorbid with other risk factors such as physical frailty, sensory impairment, social isolation, and physical dependency [13]. Other prominent factors associated with elder abuse are caregiver burnout, stress problems, coping with perpetrator childhood abuse, and relationship problems such as intimate partner violence [20].

Elder abuse is also associated with various adverse health outcomes, including physical and psychological effects that have long-term consequences such as depression, anxiety, and post-traumatic disorder [21]. It also can pose significant complications, such as premature death. For example, community-dwelling middle-aged and older women who reported prior physical, verbal, or both types of abuse had significantly higher adjusted mortality risk than women who did not report abuse [22]. In addition, reported and corroborated elder mistreatment is associated with shorter survival in both women and men [23]. Abused elders also face complications such as post-traumatic stress disorder, poorly controlled chronic diseases such as hypertension, diabetes, and heart disease, decreased quality of life, and loss of trust or quality of relationships [24]. Furthermore, elder abuse poses significant complications to society, like the cost of care for victims, the use of community legal and law enforcement resources, and the burden on nursing facilities [24].

Despite the dangers of elder abuse, information on the prevalence, types, and factors associated with elder abuse in Africa is sparse. Therefore, the present study aimed to determine the prevalence of elder abuse, the various types of abuse, and their associated factors in a Ugandan older person attending tertiary hospital outpatient clinics.

## Methods

### Study design, setting, and population

This cross-sectional descriptive study was conducted in February 2021 among older adults aged 60 years and above attending a referral hospital outpatient clinic in Uganda that attends approximately 600 older adults every month [6, 25]. After receiving care for various ailments, elders were recruited by convenience sampling after obtaining written consent from the outpatient departments. Based on the Mini-Mental Status Examination, we excluded elders with severe neurocognitive impairment (a score of 17 and below). We thus recruited 363 participants.

### Data collection

Using the list of all individuals who were attending the outpatient department, the research assistants identified participants who met the inclusion criteria of not being severely sensory impaired (deaf, dumb, or blind), ensured they got medical care first, and then obtained informed consent from them to participate in the study. Then, they collected data and interviewed each participant for a minimum of 40 minutes using translated questionnaires. The tools used in this study were translated into the local language and back-translated to ensure the meaning was maintained. However, we faced the challenges of some older adults following

instructions due to hearing or sight problems. Nonetheless, the research assistants overcame this challenge by giving more time to such older adults and speaking louder.

## Study variables and measures

**Basic information.** This study collected basic sociodemographic information including gender, age, address (rural vs. urban), occupation (previous employment, formal vs. informal; currently inactive or active at the employment), marital status, level of education (none, primary, secondary, and post-secondary education), and type of housing they are currently staying in (public/government-owned, rental, personally owned)–these show the amount of autonomy one has on their property. In addition, we asked about chronic medical illnesses such as cancer, HIV, diabetes, hypertension, etc., and physical impairments such as blindness and being orthopedically handicapped.

**Elder abuse.** The Hwalek-Sengstock Elder Abuse Screening Test (HS-EAST) was used to screen for elder abuse severity [26]. The scale consists of 15 items and three conceptual categories. These three conceptual categories include "overt violation of personal rights and direct abuse" (items 4, 9, 10, 11, 15) and "characteristics of elder that make him or her vulnerable to abuse" (items 1, 3, and 6) and characteristics of potentially abusive situations (items 2, 5, 7, 8, 12, 13 and 14) [27]. It is scored by summing responses from each item (Yes/No), where 'Yes'—scored one and 'No'—scored 0. Possible scores range from 0 to 15, where a higher score represents higher exposure to elder abuse. Based on previous studies, the HS-EAST had a cutoff of 3 and above with a sensitivity and specificity of 82.8% and 84.5%, respectively, for elder abuse; and a reported Cronbach alpha of 0.741 [28]. In this study, the Cronbach alpha was 0.78. The HS-EAST was translated into the local language and back-translated to ensure the meaning was maintained.

In addition to overall elderly abuse, different types of abuse experienced in the past six months (such as physical, emotional, and financial abuse) were assessed using questions adapted from the National Research Council on elder mistreatment monograph [19], **S1 Table**. Lastly, the older adults who preferred to report also gave information about the perpetrators.

**Functional impairment.** The level of functionality was tested using the Barthel Index (BI). The BI comprises ten items with varying weights [29]. The tool has two items assessing personal care (wash face, comb hair, shave, and clean teeth) and bathing evaluated with a 2-score scale (0 and 5 points); 6 items regarding feeding, getting onto and off the toilet, ascending and descending stairs, dressing, controlling bowels, and controlling bladder are evaluated with a 3-score t scale (0, 5, and 10 points); and two items regarding moving from wheelchair to bed and returning, and walking on a level surface are evaluated with a 4-score scale (0, 5, 10, and 15 points). The BI is a cumulative score calculated by summing each item's score. The BI scores are multiples of 5 with a range of 0 (completely dependent) to 100 (independent in basic activities of daily living (ADL)). Higher scores represent a higher degree of independence [29]. The score is categorized into five groups 0–20 = total dependency, 21–60 = severe dependency, 61–90 = moderate dependency, 91–99 = slight dependency, and 100 = complete independence. The Cronbach alpha for BI is 0.81 [29]. For this study, the Cronbach alpha was 0.88.

## Ethical considerations

Mbarara University of Science and Technology's research ethics committee approved the study (#05/11-,20), and administrative approval was obtained from the tertiary hospital director. Participants who agreed to participate in this study appended their signature or thumbprint (for those who could not read or write) on the consent form. The consent form

translated to the local language (Runyankole) was read aloud to individuals who could not read and write and signed in the presence of a trusted witness (fluent in reading and writing) who countersigned. Participants diagnosed with severe neurocognitive impairment were referred to the psychiatry department for investigation and further management. All participants were interviewed in a private room away from their caretakers and other patients to maintain privacy. Counseling was provided to all participants who experienced any form of abuse by a trained study psychologist with experience in dealing with domestic violence. Individuals who experienced abuse were referred to the district probation office–an office responsible for managing cases of violence and abuse at the district level, for further management of the situation.

## Data analysis

Data were entered into an excel sheet and then exported to STATA 16.0 for analysis. Chi-square tests were performed to determine significant differences between individuals who experienced various forms of abuse and those who did not. For elder abuse severity based on the HS-EAST, we ran a *t*-test and ANOVA for the elderly abuse severity. Linear regression analysis determined the factors associated with elder abuse severity. A back stepwise multivariable linear regression was built after testing for collinearity to adjust for confounders. In addition, logistic regression was used to determine the factors associated with the different types of elderly abuse (neglect, physical, sexual, emotional, and financial abuse). The significant level was set at less than 5% for a 95% confidence interval.

## Results

### Characteristics of the participants

A total of 363 participants were included in this study, 57% (*n* = 208) females. The mean age was 67.08 (*SD* = 6.66), and most belonged to 60–69 years (69.0%). In addition, most participants came from a rural setting (71%, *n* = 259), 72.3% had a comorbid medical condition, and 44.4% had a physical impairment (**Table 1**).

### Elder abuse

Mean elder abuse severity was 6.0 (*SD* = 2.67) out of 15 at the HS-EAST, whereas 89.0% of the elderly had met the cutoff point 3 and above the HS-EAST. In addition, these participants experienced more characteristics of potentially abusive situations, mean of 2.99 (1.47; out of 7), followed by experiencing an overt violation of personal rights and direct abuse, 2.05 (1.41; out of 5).

   Participants with physical impairment had higher elder abuse severity mean scores than those without physical impairments (6.56% vs. 5.90%, *t* = -3.71, *p*<0.001). Also, elders' who stayed in government-owned housing had higher elderly abuse severity mean scores (*F* = 4.15, *p* = 0.016), and those whose highest level of education was post-secondary had the lowest elder abuse severity score as per the education level (*F* = 5.03, *p* = 0.002) (**Table 2**).

   **Factors associated with elder abuse severity.**   At bivariate analysis, the factors that increased elder abuse severity were: being divorced or separated, previously informally employed and currently still active, having a secondary level of education, staying in a government-owned house, and having a physical impairment. These factors were tested for collinearity and had Variance Inflation Factors (VIFs) below 3; the mean VIF was 1.14. They were included in the final model using backwards stepwise modeling. There was evidence of homoskedasticity with a Cook-Weisberg test *p*-value of 0.119. The final model could explain 9.7% of elderly abuse severity. At multiple variate analyses, having a secondary level of education

**Table 1. Characteristics of the participants.**

| Variables | n (%) |
|---|---|
| **Age** | |
| 60–69 | 252 (69.0) |
| 70–79 | 90 (24.7) |
| Above 80 | 23 (6.3) |
| **Gender** | |
| Female | 208 (57.0) |
| Male | 157 (43.0) |
| **Area of dwelling** | |
| Rural | 259 (71.0) |
| Urban | 106 (29.0) |
| **Marital status** | |
| Cohabiting or married | 217 (59.4) |
| Divorced or separated | 58 (15.9) |
| Never married | 3 (0.8) |
| Widowed | 87 (23.8) |
| **Employment status** | |
| Previously formally employed, retired but currently still active | 24 (6.6) |
| Previously formally employed, retired but currently not active | 41 (11.23) |
| Previously informally employed, currently not active | 43 (11.8) |
| Previously informally employed and currently still active | 257 (70.4) |
| Level of education | |
| Never | 86 (23.6) |
| Primary | 165 (45.2) |
| Secondary | 70 (19.2) |
| Post-secondary | 44 (12.0) |
| **Type of housing** | |
| Private | 325 (89.0) |
| Public | 16 (4.4) |
| Rental | 24 (6.6) |
| **Presence of a chronic illness** | |
| No | 101 (27.7) |
| Yes | 264 (72.3) |
| **Physical impairment** | |
| No | 203 (55.6) |
| Yes | 162 (44.4) |
| **Reported perpetrators** | |
| No | 299 (81.9) |
| Yes | 66 (18.1) |
| **History of reporting abuse to police** | |
| Declines | 332 (91.0) |
| Reported | 33 (9.0) |
| **Functional dependence** | |
| Severe dependency | 11 |
| Moderate dependency | 69 |
| Slight dependency | 48 |
| Independent | 237 |

statistically significantly increased elder abuse severity by 0.97, and having a physical impairment increased the elder abuse severity by 0.83 (**Table 2**).

## Prevalence of different types of abuse

The prevalence of the different types of abuse was 86.3%, 49.0%, 46.8%, 22.2%, and 6.51, for neglect, emotional, financial, physical, and sexual abuse, respectively. About 30.4% of the participants experienced any two types of abuse (**Fig 1**), and a majority (66.6%, $n$ = 243) experienced both physical mistreatment and neglect (**Table 3**).

Experiencing physical mistreatment and emotional abuse had a low significant correlation ($r^2$ = 0.35). The elderly abuse severity significantly correlated negligibly with emotional abuse and physical mistreatment. However, the other combinations with significant correlations also had negligible correlations. For details, see **Table 4**.

**Factors associated with emotional abuse.** Previously informally employed, currently retired, and unemployed participants were more emotionally abused ($\chi^2$ = 9.08, $p$ = 0.028). Emotional abuse was less among individuals with a chronic medical condition than those with no chronic medical disease, and the difference was statistically significant (45.8% vs. 57.4%, $\chi^2$ = 3.93, $p$ = 0.047). Moderately functionally dependent elders were more emotionally abused (65.2%) than those who were more independent (43.8% and 46.0% for a slight/little dependent and completely independent, respectively) or severely functionally dependent (36.4%) ($\chi^2$ = 9.35, $p$ = 0.025) (**S2 Table**).

In bivariate analysis (**S3 Table**), the factors associated with the increase in the likelihood of experiencing emotional mistreatment were being moderately functional dependent, having to report the perpetrator to the police, or acknowledging the presence of a perpetrator. However, elders with chronic illnesses were less likely to experience emotional abuse. The factors were tested for collinearity; all had individual VIFs of less than three, and the mean VIF was 1.04. At multivariate logistic regression, the factors associated with experiencing emotional mistreatment were having reported having a perpetrator (aOR = 7.94, 95% CI: 3.71–17.06, $p$<0.001), having reported the perpetrator to the police (aOR = 3.33, 95% CI: 1.25–8.86, $p$ = 0.016), and being moderately functionally dependent (aOR = 1.85, 95% CI: 1.01–3.78, $p$ = 0.047) (**Table 5**). This final model had a specificity of 88.71%, a sensitivity of 41.90%, a positive predictive value (PPV) of 78.13%, a negative predictive value (NPV) of 61.34%, and was correctly classified as 65.75% of individuals who experienced emotional elder abuse. The goodness of fit $p$-value was 0.838 for the included four variables.

**Factors associated with physical abuse.** The largest statistical difference was between individuals who reported perpetrators and those who did not, with those who reported experiencing more physical abuse than their counterparts (50.0% vs. 16.0%; $\chi^2$ = 26.8, $p$<0.001). Older adults aged 80 years and above experienced more physical abuse than younger (43.5% vs. 22.6% and 15.6% for 60–69 years and 70–79 years, respectively) ($\chi^2$ = 8.36, $p$ = 0.015). All individuals who never married experienced physical abuse, whereas 21.7%, 22.4%, and 20.7% of individuals who were either cohabiting/married, divorced/separated, or widowed, respectively, experienced physical abuse ($\chi^2$ = 10.67, $p$ = 0.014). Older adults having a chronic medical illness were statistically less physically abused (17.4%) compared to those with no chronic medical condition (34.6%) ($\chi^2$ = 12.6, p-value <0.001). Older adults with moderate functional dependence were the main culprit for physical abuse (37.7%) compared with individuals who were more functionally independent (13.6% and 19.8% for slightly dependent and completely independent, respectively) ($\chi^2$ = 13.05, $p$ = 0.005) (**S2 Table**).

At bivariate analysis, individuals with chronic medical illnesses were less likely to experience physical abuse (**S3 Table**). However, those above 80 years had reported the perpetrator to

**Table 2. Associations between elder abuse severity and the studied variables.**

| Variables | Elderly abuse severity | | Bivariate linear regression | | Multivariate linear regression | |
|---|---|---|---|---|---|---|
| | Mean (SD) | F(p-value) | coefficient (standard beta) | p-value | coefficient (standard beta) | p-value |
| **Age** | | | | | | |
| 60–69 | 6.02 (2.74) | 0.84 (0.614) | 1 | | | |
| 70–79 | 6.09 (2.57) | | 0.07 (0.01) | 0.834 | | |
| Above 80 | 5.70 (2.30) | | -0.32 (-0.03) | 0.578 | | |
| **Gender** | | | | | | |
| Female | 6.23 (2.66) | 1.73 (0.084) | 1 | | | |
| Male | 5.74 (2.67) | | -0.49 (-0.09) | 0.084 | | |
| **Area of dwelling** | | | | | | |
| Rural | 5.99 (2.70) | -0.27 (0.797) | 1 | | | |
| Urban | 6.07 (2.59) | | 0.08 (0.01) | 0.787 | | |
| **Marital status** | | | | | | |
| Cohabiting or married | 5.79 (2.57) | 2.20 (0.088) | 1 | | 1 | |
| Divorced or separated | 6.79 (2.38) | | 1.00 (0.14) | 0.011 | 0.74 (0.10) | 0.061 |
| Never married | 5.67 (2.89) | | -0.13 (-0.01) | 0.935 | -0.95 (-0.03) | 0.538 |
| Widowed | 6.07 (2.58) | | 0.28 (0.04) | 0.413 | 0.26 (0.04) | 0.434 |
| **Employment status** | | | | | | |
| Previously formally employed, retired but currently still active | 4.96 (2.53) | 2.06 (0.105) | 1 | | 1 | |
| Previously formally employed, retired but currently not active | 5.71 (3.11) | | 0.75 (0.09) | 0.273 | 0.35 (0.04) | 0.603 |
| Previously informally employed, currently not active | 5.74 (2.29) | | 0.79 (0.09) | 0.246 | 0.16 (0.02) | 0.823 |
| Previously informally employed and currently still active | 6.21 (2.65) | | 1.25 (0.21) | 0.028 | 0.59 (0.10) | 0.363 |
| **Level of education** | | | | | | |
| Never | 5.64 (2.63) | 5.03 (0.002) | 1 | | 1 | |
| Primary | 6.21 (2.69) | | 0.57 (0.11) | 0.106 | 0.63 (0.12) | 0.072 |
| Secondary | 6.71 (2.54) | | 1.07 (0.16) | 0.011 | 0.97 (0.14) | 0.028 |
| Post-secondary | 4.93 (2.50) | | -0.71 (-0.09) | 0.147 | -0.31 (-0.04) | 0.606 |
| **Type of housing** | | | | | | |
| Private | 5.90 (2.67) | 4.15 (0.016) | 1 | | 1 | |
| Public | 7.81 (2.10) | | 1.91 (0.15) | 0.005 | 1.08 (0.08) | 0.121 |
| Rental | 6.33 (2.66) | | 0.43 (0.04) | 0.444 | 0.40 (0.04) | 0.476 |
| **Presence of a chronic illness** | | | | | | |
| No | 5.98 (2.71) | -0.16 (0.873) | 1 | | | |
| Yes | 6.03 (2.66) | | 0.05 (0.01) | 0.873 | | |
| **Physical impairment** | | | | | | |
| No | 5.56 (2.72) | -3.71 (<0.001) | 1 | | 1 | |
| Yes | 6.59 (2.50) | | 1.02 (0.19) | <0.001 | 0.83 (0.15) | 0.003 |
| **Reported perpetrators** | | | | | | |
| No | 5.90 (2.61) | -1.84 (0.067) | 1 | | | |
| Yes | 6.56 (2.89) | | 0.66 (0.96) | 0.067 | | |
| **History of reporting abuse to police** | | | | | | |
| Declines | 5.93 (2.63) | | 1 | | | |
| Reported | 6.88 (2.92) | -1.95 (0.051) | 0.95 (0.10) | 0.051 | | |
| **Functional dependence** | | | | | | |

(*Continued*)

**Table 2.** (Continued)

| Variables | Elderly abuse severity | | Bivariate linear regression | | Multivariate linear regression | |
|---|---|---|---|---|---|---|
| | Mean (SD) | F(p-value) | coefficient (standard beta) | p-value | coefficient (standard beta) | p-value |
| Severe dependency | 5.18 (2.75) | 0.62 (0.600) | 1 | | | |
| Moderate dependency | 5.97 (2.29) | | 0.79 (0.11) | 0.364 | | |
| Slight dependency | 6.35 (2.85) | | 1.17 (0.15) | 0.190 | | |
| Independent | 6.00 (2.73) | | 0.82 (0.15) | 0.322 | | |

the police or acknowledged their presence, and being moderately functionally dependent increased the likelihood of experiencing physical mistreatment at bivariate analysis. These factors were tested for collinearity, and they all had VIFs below 3, with a mean VIF of 1.02, and consequently, they were included in the final model. The model had a sensitivity of 28.40%, specificity of 95.07%, a PPV of 62.16%, and NPV of 82.32%, and correctly classified 80.27% of experiencing physical mistreatment. The goodness-of-fit p-value was 0.639 and included five variables. In the multivariable logistic regression, the factors associated with experiencing abuse were having reported the presence of a perpetrator (aOR = 5.12, 95% CI: 2.79–9.44, $p < 0.001$) and an elderly who has moderate functional dependence (aOR = 2.44, 95% CI: 1.13–4.08, $p = 0.019$). However, chronic physical medical conditions reduced the likelihood of experiencing physical mistreatment (aOR = 0.40, 95% CI: 0.23–0.71, $p = 0.002$) (**Table 5**).

**Factors associated with neglect.** There was a statistical difference between neglect and functional dependence ($\chi^2 = 18.78$, $p < 0.001$); neglect was more among elders with higher levels of dependence. Participants with physical impairment were statistically less neglected than those without physical impairment (79.0% vs. 92.1%, $\chi^2 = 13.09$, $p < 0.001$). Elders who reported having a perpetrator (95.4%) were statistically neglected more compared to those who declined (84.3%) ($\chi^2 = 5.71$, $p = 0.017$) (**S2 Table**).

In bivariate analysis (**S3 Table**), neglect was associated with having the reported presence of a perpetrator and an increasing level of functional dependence; that is, those with moderate functional dependence have a higher likelihood of experiencing neglect than those with slight functional dependence. Staying in publicly owned/government housing and having a physical impairment significantly reduced the likelihood of experiencing neglect. These factors had a mean VIF of 1.01, and all their VIFs were below three at testing for collinearity. In multivariate analysis (**Table 5**), having physical impairment reduced the likelihood of neglect (aOR = 0.27, 95% CI: 0.14–0.52, $p < 0.001$). However, individuals who reported the presence of a perpetrator and those with slight or moderate functional dependence were likely to experience neglect. The likelihood of experiencing neglect was highest in individuals with moderate functional dependence (aOR = 9.33, 95% CI: 2.15–40.42, $p = 0.003$). This final model had a sensitivity of 100%, specificity of 0%, PPV of 85.88%, and correctly classified 85.88% of neglect experienced by older adults.

**Factors associated with financial abuse.** There was a statistical difference between financial abuse and the following variables: age, gender, dwelling area, employment status, level of education, presence of chronic illness, presence of physical impairment, history of a police report, and functional dependence. Participants aged 80 and above were statistically more financially abused (91.3%) compared with younger participants (43.2% and 45.6% for 60–69 and 70–79, respectively) ($\chi^2 = 19.6$ $p < 0.001$). Male participants suffered more financial abuse than females (54.8% vs. 40.9%; $\chi^2 = 6.95$, $p = 0.008$). Urban dwellers statistically suffered more elderly abuse than rural dwellers (55.7% vs. 43.2%; $\chi^2 = 4.66$, $p = 0.031$). For details, see **S2 Table**.

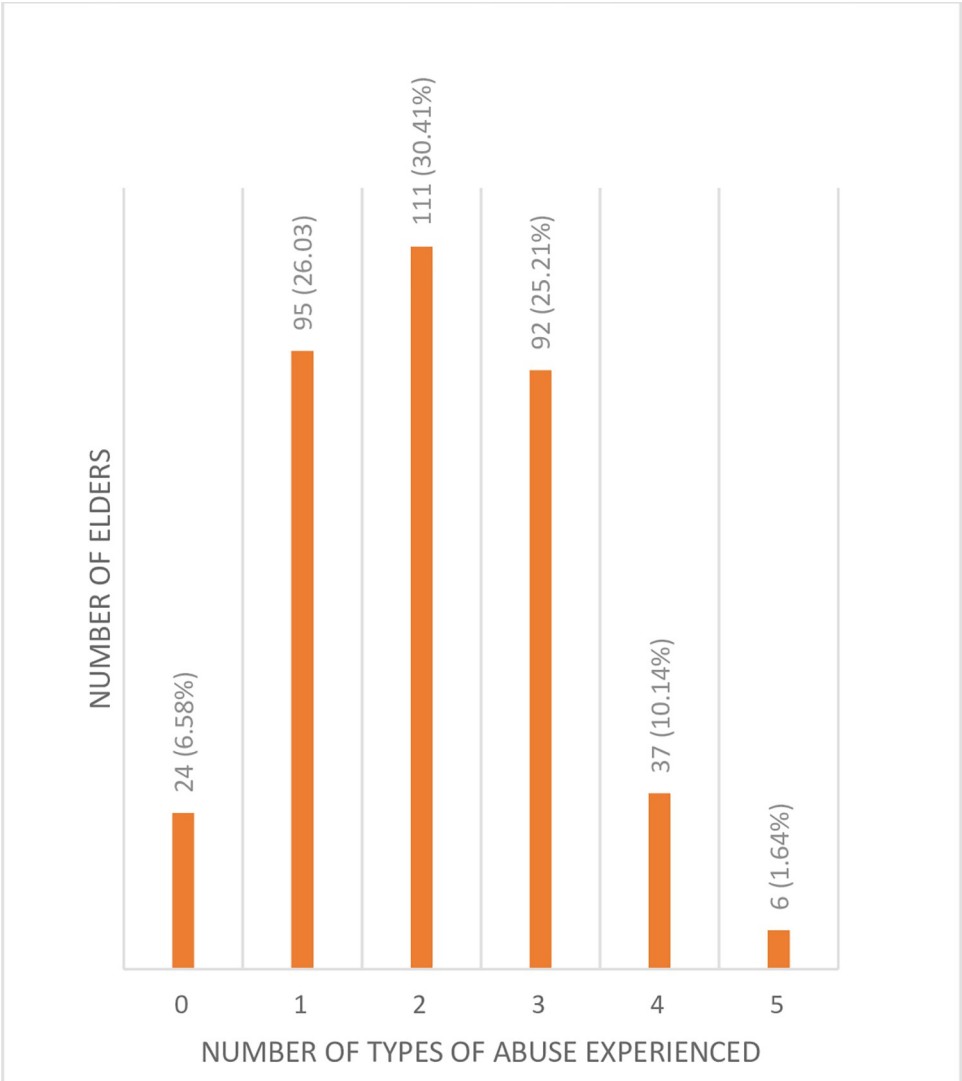

**Fig 1. Histogram of number of abuses experienced by the elders.**

In bivariate analysis (**S3 Table**), having experienced financial abuse was associated with staying in an urban setting, male gender, age above 80 years, level of education (primary and tertiary), and severe functional dependence. The likelihood was reduced among individuals with chronic physical medical conditions, those with physical impairment, and those who reported their perpetrators to the police. The factors significant at bivariate analysis were tested

**Table 3. Participants experiencing two types of abuse.**

| Type of abuse | 1 | 2 | 3 | 4 | 5 |
|---|---|---|---|---|---|
| Sexual (1) | - | | | | |
| Emotional (2) | 8 | - | | | |
| Physical mistreatment (3) | 14 | 113 | - | | |
| Neglect (4) | 5 | 15 | 243 | | |
| Financial (5) | 11 | 83 | 28 | 159 | - |

**Table 4. Correlation between the different types of abuse and elderly abuse severity.**

| Type of abuse | 1 | 2 | 3 | 4 | 5 | 6 |
|---|---|---|---|---|---|---|
| Sexual (1) | 1 | | | | | |
| Emotional (2) | **0.10**[*] | 1 | | | | |
| Physical mistreatment (3) | **0.14**[**] | **0.35**[**] | 1 | | | |
| Neglect (4) | -0.05 | **0.15**[**] | 0.04 | 1 | | |
| Financial (5) | 0.05 | **0.13**[*] | **0.20**[**] | **0.13**[**] | 1 | |
| Elderly abuse severity (6) | 0.03 | **0.17**[**] | **0.17**[**] | -0.11 | -0.48 | 1 |

[*] = p value less than 0.05

[**] = p value less than 0.001; very high correlation positive (negative) = $r^2$ 0.90 to 1.00 (−0.90 to −1.00); high positive (negative) correlation = 0.70 to 0.90 (−0.70 to −0.90); moderate positive (negative) correlation = 0.50 to 0.70 (−0.50 to −0.70); low positive (negative) correlation = 0.30 to 0.50 (−.30 to −0.50); Negligible correlation = 0.00 to 0.30 (.00 to −0.30).

for collinearity; all had a VIF of less than 3 with a mean VIF of 1.08. All were included in the final multivariate logistic model whose sensitivity was 57.31%, specificity of 76.29%, PPV of 68.06%, NPV of 66.97%, and correctly classified 67.40% of financial abuse. Physical impairment and reporting the perpetrator to the police were significant factors that reduced the likelihood of financial abuse. However, age above 80 (aOR = 4.24, 95% CI: 2.37–53.21, p = 0.002), having primary (aOR = 2.01 95% CI: 1.08–3.72, p = 0.0027) or secondary (aOR = 3.65, 95% CI: 1.48–9.02, p = 0.005) as the highest level of education, and being severely functional dependent (aOR = 8.95, 95% CI: 1.57–51.08, p = 0.014) significantly increased the likelihood of experiencing financial abuse (**Table 5**).

**Factors associated with sexual abuse.** There was no gender difference between the level of sexual abuse experienced (**S2 Table**). No factor was associated with sexual abuse (**S3 Table**).

## Discussion

In this survey of older adults aged 60 years and above attending an outpatient department of a tertiary hospital in Southwestern Uganda, the prevalence of elder abuse (cutoff of 3 out of 15 at the HS-EAST) was high (89.0%). The factors associated with increased elder abuse severity were having a secondary level of education and physical impairment. However, the prevalence of the various types of elder abuse was highest with neglect (86.0%) and lowest with sexual abuse (6.8%), and about 30.4% of the abused experienced at least two forms of abuse, especially physical mistreatment and neglect. Moderate to severe functional dependence was associated with all forms of abuse apart from sexual abuse. Individuals who reported a perpetrator were likely to experience neglect, emotional abuse, and physical mistreatment. However, those who reported their perpetrators to the police had a higher likelihood of experiencing emotional abuse but were less likely to experience financial abuse. Emotional abuse was also associated with age above 80 years and attaining education (primary and secondary). Physical impairment and chronic medical conditions reduced the likelihood of experiencing neglect and financial abuse, and physical abuse, respectively.

### Elder abuse

The prevalence of elderly abuse (89.0%) was much higher than that estimated from India (50.2%) [30], the USA (35.0%) [31], China (36.2%) [32, 33], and Iran (38.5%) [34] that used a similar tool—the HS-EAST. In addition it is higher than that estimated in other rural areas [14]. The high prevalence in this study may be because we recruited a community sample of older persons attending a hospital outpatient clinic and some of the victims visited the hospital

**Table 5. Multivariable logistic regression analysis for factors associated with the different types of abuse.**

| Variables | Neglect | | Financial abuse | | Emotional mistreatment | | Physical mistreatment | |
|---|---|---|---|---|---|---|---|---|
| | aOR (95% CI) | p-value | aOR (95% CI) | p-value | aOR (95% CI) | p-value | aOR (95% CI) | p-value |
| Age | | | | | | | | |
| 60–69 | | | 1 | | | | 1 | |
| 70–79 | | | 1.15 (0.67–1.99) | 0.606 | | | 0.71 (0.35–1.43) | 0.337 |
| Above 80 | | | 11.24 (2.37–53.21) | 0.002 | | | 1.83 (0.68–4.88) | 0.227 |
| Gender | | | | | | | | |
| Female | | | 1 | | | | | |
| Male | | | 1.25 (0.77–2.03) | 0.367 | | | | |
| Area of dwelling | | | | | | | | |
| Rural | | | 1 | | | | | |
| Urban | | | 1.37 (0.80–2.35) | 0.249 | | | | |
| Level of education | | | | | | | | |
| Never | | | 1 | | | | | |
| Primary | | | 2.01 (1.08–3.72) | 0.027 | | | | |
| Secondary | | | 1.27 (0.61–2.66) | 0.523 | | | | |
| Tertiary | | | 3.65 (1.48–9.02) | 0.005 | | | | |
| Type of housing | | | | | | | | |
| Private | 1 | | | | | | | |
| Public | 0.60 (0.18–1.98) | 0.397 | | | | | | |
| Rental | 0.83 (0.22–3.15) | 0.783 | | | | | | |
| Presence of a chronic illness | | | | | | | | |
| No | | | 1 | | 1 | | 1 | |
| Yes | | | 0.75 (0.43–1.29) | 0.296 | 0.61 (0.37–1.01) | 0.053 | 0.40 (0.23–0.71) | 0.002 |
| Physical impairment | | | | | | | | |
| No | 1 | | 1 | | | | | |
| Yes | 0.27 (0.14–0.52) | <0.001 | 0.40 (0.24–0.65) | <0.001 | | | | |
| Reported perpetrators | | | | | | | | |
| No | 1 | | | | 1 | | 1 | |
| Yes | 3.49 (1.01–12.03) | 0.047 | | | 7.94 (3.71–17.06) | <0.001 | 5.14 (2.79–9.44) | <0.001 |
| History of reporting abuse to police | | | | | | | | |
| No | | | 1 | | 1 | | | |
| Yes | | | 0.31 (0.12–0.75) | 0.009 | 3.33 (1.25–8.86) | 0.016 | | |
| Functional dependence | | | | | | | | |
| Severe dependency | Omitted | | 8.95 (1.57–51.08) | 0.014 | 0.52 (0.12–2.22) | 0.533 | 0.32 (0.04–2.89) | 0.312 |
| Moderate dependency | 9.33 (2.15–40.42) | 0.003 | 1.73 (0.92–3.26) | 0.090 | 1.85 (1.01–3.78) | 0.047 | 2.44 (1.13–4.08) | 0.019 |
| Slight dependency | 6.48 (1.48–28.34) | 0.013 | 1.01 (0.50–2.03) | 0.980 | 1.04 (0.53–2.05) | 0.898 | 0.78 (0.31–1.95) | 0.597 |
| Independent | 1 | | 1 | | 1 | | 1 | |

to treat complications of elder abuse such as injuries, illness, and mental health challenges [35]. In addition, the current study was in a country with limited laws, reporting, and follow-up of elderly abuse, on top of a cultural system diluted to the extent of the young or their children disrespecting the elderly [5].

In this study, having a secondary level of education (more than eight years of study in Uganda) increased the elder abuse severity. This is contradictory to other study findings, which report that a history of education for more than eight years reduced elder abuse severity [36]. Older adults with higher education levels can claim their rights, easily report abuse to responsible bodies, and even get more social respect [36]. However, recently older adults in

Uganda have been disrespected by the young due to rapid cultural change [5], which puts many older adults at risk of abuse. In addition, the country is a low-income country with challenges such as unemployment, poverty, and a high fertility rate—many of the younger individuals financially exploit the weak and vulnerable elderly, including the educated. In this country, the educated are more likely to be financially stable, having accumulated some assets over time or are receiving a pension, and therefore become prey to the younger generation (their dependents) for their assets and money [5]. No wonder these older adults with a higher level of education were at a higher likelihood of being financially abused in our study. Furthermore, older adults with higher levels of education are most likely to have had formal jobs in the city and lost contact with extended family ties. Therefore, they have poor relationships with their relatives when they return to the villages after retirement, yet these ties could have been a buffer from abuse.

Also, older adults with physical impairment had more severe elder abuse in this study, a finding similar to that reported in the previous studies [17, 32]. This finding is not surprising as people with disabilities depend on others, especially in fulfilling their daily activities [37]. In addition, older adults with physical impairment may not defend themselves or report their perpetrators to the police due to their inability to fight for themselves and fear of being neglected.

## Financial abuse

Almost half of the participants experienced financial abuse, which was higher than reported in other African countries, including Nigeria (13.1%) [38], Egypt (27%) [39], and South Africa (24.4%) [17]. Despite all these African countries belonging to the LMICs, Uganda has one of the lowest Gross Domestic Product [40]; thus, its citizens experience a higher poverty level, leading them to abuse financially stable or dependent older adults. Financially stable or dependent older adults become a primary target by some poor youth due to their vulnerabilities due to old age. Financial abuse in this study was associated with increasing age above 80 years because individuals above 80 years are more vulnerable to abuse. After all, they are at higher risk of neurocognitive disorders and functional dependence, which makes them unable to manage their finances or forget to request financial assistance [6]. They may also be more likely to make poor financial decisions and fall victim to fraudulent deals [41]. An increase in functional dependency with age is a major factor that increases the likelihood of almost all types of abuse except sexual abuse due to increased vulnerability and need for help in most of their daily activities. An elderly with a high level of functional dependence causes significant care burden stress, leading to an increased likelihood of abuse [42].

The likelihood of being financially abused was reduced when the older adult reported the known victim to the police. Reporting the perpetrators to the police, a method suggested by the CDC to reduce abuse, was effective in this study [4]. A perpetrator reported to the police may be punished for their acts, for instance, by being taken away from the community or the victims' premises, thus, reducing further abuse. In addition, reporting a perpetrator to the police will make the community aware of the perpetrating persons, which will protect the potential victims. This community support may provide emotional and social support, thus reducing the likelihood of experiencing emotional abuse and other forms of elder abuse. In Uganda, the community considers abuse of an older adult taboo [5]. Disrespecting or abusing an older adult with a disability is considered a worse offense by the community. Due to this respect and possible fear of the community reaction, many perpetrators do not abuse older adults with physical disabilities. The respect they give individuals with a disability may be responsible for them being less likely to experience financial abuse and neglect since every community member expects one to take good care of them.

## Physical abuse and mistreatment

The prevalence of physical abuse (22.2%) among the older adults in this study was higher than that reported by most countries in a systematic review of 20 studies (0.2%–4.9%) [43]. However, studies included in the review by Pillemer et al. (2016) measured physical abuse using the conflict tactic scale that predominantly looks at intimate partner violence, excluding abuse from other possible perpetrators such as children, neighbors, and others; considered in our study. Thus, a large difference between the reported prevalence rates. On the other hand, the current study's prevalence of physical abuse was less than that reported by a Nigerian study (47.0%) [44]. The difference may be due to cultural belief differences between the two countries and associating older adults with witchcraft due to their appearance (wrinkled skin, gnarled hands, and yellow eyes) [45]. In addition, the Nigerian study classified physical neglect as part of physical abuse, which could have led to a higher prevalence since neglect is a commonly reported type of elder abuse [46].

Like other types of elder abuse, experiencing physical abuse was associated with increased functional dependence. The individuals who reported the presence of perpetrators had a higher likelihood of physical abuse; this may be due to the demeaning nature of physical abuse and the desire to get help from others (i.e., reporting is an act of asking for help). However, individuals with chronic physical medical conditions were less likely to experience physical abuse since they looked too sick. Their perpetrators may not abuse them due to fear of them possibly killing them. It contradicted other studies where chronic diseases were associated with physical abuse [47].

## Emotional abuse

Nearly half of all older adults in this study experienced emotional abuse, comparable to findings in South Africa, Nigeria, and Egypt [17, 39, 48]. These findings could be because there are ways of addressing older adults in traditional African society, and a deviation from this norm is easily noticeable and considered disrespectful [38]. However, the prevalence of emotional abuse was much higher than that reported by Cadmus and Owoaje (2012) in Nigeria among women without psychiatric illnesses [38]. Despite psychiatric illnesses such as depression being associated with or a complication of emotional abuse [21]. The psychological suffering associated with emotional abuse was associated with reporting the presence of a perpetrator to the research team and police in an attempt to seek emotional relief.

## Neglect

As reported by other researchers in Africa [30, 39], neglect was the most dominant type of abuse. This high prevalence could be attributed to the loss of caregivers due to the HIV scourge in Africa or the migration of family members to urban areas compounded by the loss of extended family ties leaving the elders to fend for themselves and hence feel neglected [49]. Poverty is another factor that could explain high-rate neglect, as caregivers cannot take care of their own families, let alone older adults [50]. In addition, this study recruited older adults seeking care in hospitals with a high probability of having chronic illnesses, which put them at an increased likelihood of neglect due to limitations in activities of daily living and their associated high financial demands from the caregivers [20, 50, 51]. On the other hand, participants with physical impairment had a reduced likelihood of being neglected. Since a physical impairment limits an older adult's ability to manage daily life activities, putting them at risk of self-neglect [52, 53], this vulnerability makes caregivers present. The caregivers take care of them since not caring for the physically impaired is not culturally accepted [5].

## Sexual abuse

Sexual abuse was the least prevalent type, similar to other studies [15, 17, 38, 46, 54, 55]. In the current study, there was no factor associated with sexual abuse. However, studies in South Africa and Canada have reported that women are at a higher risk of sexual abuse than males [17, 55]. In Africa, sexual abuse of older persons seldom occurs, and when it occurs, it may be a ritual related to getting rich or getting special spiritual powers [56, 57]. In addition, sexual abuse of older adult women is not expected as ageist perceptions produce a taboo around considering sexual relations with older people [58]. Furthermore, older persons who are sexually abused may not report sexual abuse due to feelings of shame [59].

## Limitations

First, the study was cross-sectional in design, so the causality between elder abuse and its predictors cannot be determined. Second, the study involved older persons attending a hospital outpatient for the management of their illnesses; therefore, elders at higher risk of abuse were sampled, leading to overestimation the prevalence rates. Second, elders with severe cognitive impairment were not sampled and yet are at higher risk of abuse. This sample may have difficulty with the recall of abuse and may end up inflating the number of cases of abuse reported, thus, excluded in the present study. However, we recommend future studies use prospective methodology involving individuals with severe cognitive impairment and examine the participants routinely for abuse; to enable accurate estimation of abuse in this population. In addition, neurocognitive scores were not considered in the analysis despite decreased scores being associated with abuse even without severe impairment. Thirdly, some variables, such as family income or socioeconomic factors, were not explored in detail despite their role in elderly abuse based on previous studies. Older people with lower socioeconomic independence were reported to have more abuse [60]. We recommend future studies add variables to explore the socioeconomic status of the elder, such as family income, saving used by the older person, income sources, and the number of dependents, among others. Lastly, the tool used to determine abuse was not adopted in Uganda; therefore, it could have been estimated as higher than the actual prevalence.

## Conclusions

This study highlights a high prevalence of elder abuse in Uganda, with the most common forms of abuse being neglect, emotional abuse, and financial abuse. In addition, our study showed that having at least a secondary level of education or physical impairment was associated with elder abuse. Older persons with physical impairment were less likely to be neglected, and those who reported perpetrators were more likely to be abused emotionally. Older persons above 80 years were more likely to be abused financially, but those who reported perpetrators to the police or had a physical disability were less likely to face financial abuse. Furthermore, older persons with increased functional dependence were more likely to be abused physically, but those with chronic illness were less likely to be abused. This study, therefore, is evidence of the need to implement more effective means of raising awareness about elder abuse. There is also a need to design interventions to prevent vice in these vulnerable groups, such as routine screening for abuse at health facilities where older persons have access to care. However, this may be difficult due to few personnel in health centers to screen older adults in addition to treating their health conditions in rural areas. Policies that address older person abuse should be put in place. Lastly, we recommend further studies on elder abuse in this setting, especially on its psychological impact and among the neurocognitively impaired, since these aspects were not exploited in this study.

## Supporting information

**S1 Data.**
(RAR)

**S1 Table. Elder abuse questionnaire.**
(DOCX)

**S2 Table. Relationship between the type of abuse and study variable.**
(DOCX)

**S3 Table. Bivariate logistic regression analysis for factors associated with the different types of abuse.**
(DOCX)

## Acknowledgments

We acknowledge the research assistants Franklin Kakuru, Brenda Nabatanza, Badru Kayongo, Innocent Arinaitwe, and Sarah Maria Najjuka. Mbarara regional referral hospital provided a conducive environment for data collection, and without the participants, this data would not have been obtained.

## Author Contributions

**Conceptualization:** Letizia Maria Atim, Mark Mohan Kaggwa, Scholastic Ashaba, Samuel Maling.

**Data curation:** Letizia Maria Atim, Mark Mohan Kaggwa, Samuel Maling.

**Formal analysis:** Letizia Maria Atim, Mark Mohan Kaggwa.

**Funding acquisition:** Letizia Maria Atim, Mark Mohan Kaggwa, Samuel Maling.

**Investigation:** Letizia Maria Atim, Mark Mohan Kaggwa, Moses Kule, Scholastic Ashaba, Samuel Maling.

**Methodology:** Letizia Maria Atim, Mark Mohan Kaggwa, Scholastic Ashaba, Samuel Maling.

**Project administration:** Letizia Maria Atim, Mark Mohan Kaggwa, Moses Kule, Samuel Maling.

**Resources:** Letizia Maria Atim, Mark Mohan Kaggwa, Samuel Maling.

**Software:** Mark Mohan Kaggwa.

**Supervision:** Letizia Maria Atim, Mark Mohan Kaggwa, Scholastic Ashaba, Samuel Maling.

**Validation:** Letizia Maria Atim, Mark Mohan Kaggwa, Samuel Maling.

**Visualization:** Letizia Maria Atim, Mark Mohan Kaggwa, Mohammed A. Mamum, Scholastic Ashaba, Samuel Maling.

**Writing – original draft:** Letizia Maria Atim, Mark Mohan Kaggwa.

**Writing – review & editing:** Letizia Maria Atim, Mark Mohan Kaggwa, Mohammed A. Mamum, Moses Kule, Scholastic Ashaba, Samuel Maling.

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
