## [Decision Letter · Decision Letter 0]

26 Oct 2022

PONE-D-22-24400Factors associated with elder abuse in rural Uganda: A hospital-based cross-sectional studyPLOS ONE

Dear Dr. Kaggwa,

Thank you for submitting your manuscript to PLOS ONE. After careful consideration, we feel that it has merit but does not fully meet PLOS ONE’s publication criteria as it currently stands. Therefore, we invite you to submit a revised version of the manuscript that addresses the points raised during the review process.

I agree with the reviewers on how necessary and important it is to carry out studies like this and I thank the authors for having this type of interest in the elderly population, previously always respected and today so frequently mistreated. I encourage authors to follow the suggestions made by reviewers in an effort to get their manuscript ready for publication.

We look forward to receiving your revised manuscript.

Kind regards,

Thalia Fernandez, Ph.D.

Academic Editor

PLOS ONE

Journal Requirements:

Reviewers' comments:

Reviewer's Responses to Questions

**Comments to the Author**

1. Is the manuscript technically sound, and do the data support the conclusions?

Reviewer #1: Yes

Reviewer #2: Yes

2. Has the statistical analysis been performed appropriately and rigorously? 

Reviewer #1: Yes

Reviewer #2: Yes

3. Have the authors made all data underlying the findings in their manuscript fully available?

Reviewer #1: Yes

Reviewer #2: Yes

4. Is the manuscript presented in an intelligible fashion and written in standard English?

Reviewer #1: Yes

Reviewer #2: Yes

5. Review Comments to the Author

Reviewer #1: Thank you for doing this study. The topic is important, along with your findings. Please see my comments:

Pg8 line 29: The population is actually community older adults who attend an hospital based outpatient clinic, therefore they are not hospital population per se.

Pg9 line 62: What is LMIC? Please define this term before using an abbreviation for it.

Pg10 lines 115-117: How were they recruited exactly? How was consent formally obtained?

Pg10 lines 119-120: What do you mean 'fast tracked them to recieve medical care'?

Pg10 line 130: What do you mean by the terms 'lame' and 'crippled'?

Pg11 line 145: What do you exactly mean by this, and what information did they give exactly

Pg11 lines 146-159: Did you look at higher levels of functionality, such as the ability to complete the instrumental acitivities of daily living, like managing ones own finances? If not, why not? This is particularly important when you are considering financial abuse.

Pg12 line 207: Please define what a VIF is.

Pg16 line 337: But this was not a hospital sample, it was from a sample derived from an outpatient clinic based in a hospital setting

Pg16 lines 344-351: I am not sure this adequately explains your findings - are there any reaons to explain your contradictory findings?

Pg18 lines 445-449: Additional limitations include the exclusion of people with more severe neurocognitive disorders, whom are at higher risk of abuse

Pg18 line 446: See my comment earlier about this being based in the Outpatient setting, not the hospital

Pg18 Lines 451-455: Please expand more on this section, including a summary of the findings, and future opportunities

Reviewer #2: This paper examines the prevalence of elder abuse in a hospital-based sample of adults from rural Uganda.

Given the social and medical significance of elder abuse, the general lack of awareness of the severity of this problem, and the paucity of literature on the subject from this region, the current study is meritorious from a public health perspective. I appreciate the authors’ work, as well as their thorough coverage of the ethical complexities involved in such research.

The following are aspects of the paper which would benefit from correction or clarification:

1. Abstract:

a. It is stated that about one in ten elderly individuals experiences abuse. This estimate is somewhat on the lower side; a recent systematic review of elderly people in rural areas suggests that the figure is closer to 33% (i.e., one in three) – cf. Zhang et al., 2022. This could be reflected in both the abstract and the text, as this study is based on a sample of elderly people from a rural area.

b. Has this study examined only abuse, neglect, or both abuse and neglect? The title and abstract refer to abuse but there is a mention of neglect in line 43.

2. Introduction:

a. The authors could incorporate recent systematic reviews of elder abuse, particularly from rural or non-Western countries, into this part of the text. See Zhang et al. (mentioned above), Wang et al. (2022) or Ho et al. (2017).

b. As this study is being conducted in rural Uganda, it would be informative to discuss those unique aspects of Ugandan culture that could either contribute to or protect against elder abuse. (For example, if respect for elders is a national cultural value, this could act as a protective factor; on the other hand, if there is a “culture of silence” about violence against women, as in many Asian and African cultures including this reviewer’s, this could be a risk factor for elderly women specifically.)

3. Methods:

a. Was the sample size of n = 363 sufficient to address the research questions? Were any subjects excluded based on criteria other than neurocognitive impairment (e.g., refusal of consent by the individual or by their family members?)

b. Terms such as “lame” and “crippled”, though vivid and descriptive, may be considered pejorative or “politically incorrect” by many modern readers. It is preferable if the authors use medical terms (e.g., orthopedically handicapped).

c. Did the instrument used to measure elder abuse require any modifications / adaptations based on local culture or idioms? If so, this could be mentioned. Did any of the study participants find it difficult to understand some of the concepts covered by this instrument?

d. Likewise, if the study instruments were translated into a local language for the purpose of this study, details of translation and validation should be provided in this section.

e. Were any practical difficulties faced by the researchers in the course of interviewing each subject for 40 minutes (e.g., hearing impairment, fatigue, emotional lability, catastrophic reactions)?

f. Apart from measures of functional capacity, did the authors collect any data on the mental health of the participants (e.g., symptoms of depression or anxiety)? This is important because there is a close relationship between elder abuse and mental health.

g. Were the MMSE scores also included in the data analysis? (There is some evidence that cognitive impairment, even if not in the severe range, could be a risk factor for elder abuse – see Lachs et al., 1997)

h. Was any information collected on family income / socioeconomic status? This could also be a potential risk factor.

4. Results:

a. See 3 g. above; if not done, this can be mentioned as a limitation / direction for future research.

b. Was information available on the different types / symptoms of physical illness in relation to elder abuse (e.g., were specific illnesses associated with visual or hearing impairment, incontinence, etc. associated with a higher risk of abuse)?

c. Were any meaningful differences in predictor variables for abuse / neglect identified when comparing elderly men and women (i.e., were there gender differences in risk factors)?

5. Discussion:

This section should be more than a repetition of the study results. For example, the authors could compare their finding of a prevalence of 89% with other studies from rural areas, which yield an estimate of 4 – 61%, and explore the possible reasons (methodological as well as actual) which could account for such a high estimate. The authors could also discuss recommendations for the prevention and management of elder abuse based on their study results (e.g., based on the predictor variables identified in the multivariate analyses) as well as the limitations / difficulties associated with managing this problem in a rural Ugandan setting.

6. PLOS authors have the option to publish the peer review history of their article (what does this mean?). If published, this will include your full peer review and any attached files.

Reviewer #1: **Yes: **Frances Carr

Reviewer #2: **Yes: **Ravi Philip Rajkumar

---

## [Author Response · Author response to Decision Letter 0]

14 Nov 2022

Response to the Journal Requirements 

Journal Requirement: Please ensure that your manuscript meets PLOS ONE's style requirements, including those for file naming. The PLOS ONE style templates can be found at https://journals.plos.org/plosone/s/file?id=wjVg/PLOSOne_formatting_sample_main_body.pdf and 

Author’s Response: The manuscript has been re-arranged as suggested. Thank you.

Journal Requirement: Your ethics statement should only appear in the Methods section of your manuscript. If your ethics statement is written in any section besides the Methods, please delete it from any other section.

Author’s Response: The manuscript has been rectified as suggested on page 10. Thank you.

Journal Requirement: Please review your reference list to ensure that it is complete and correct. If you have cited papers that have been retracted, please include the rationale for doing so in the manuscript text, or remove these references and replace them with relevant current references. Any changes to the reference list should be mentioned in the rebuttal letter that accompanies your revised manuscript. If you need to cite a retracted article, indicate the article’s retracted status in the References list and also include a citation and full reference for the retraction notice.

Author’s Response: This has been followed as instructed. Thank you.

Response to Reviewer 1’s Comments

Comment 1: Pg8 line 29: The population is actually community older adults who attend a hospital-based outpatient clinic; therefore, they are not a hospital population per se.

Response: Thank you. We have rectified the text as suggested. 

Comment2: Pg9 line 62: What is LMIC? Please define this term before using an abbreviation for it.

Response: Thank you for the comment. This abbreviation has been defined before use on page 4 line 57. 

Comment 3: Pg10 lines 115-117: How were they recruited exactly? 

Response: We have clearly described the recruiting process in the data collection section on page 7 lines 125-133 It is described as: 

“Using the list of all individuals who were attending the outpatient department, the research assistants identified participants who met the inclusion criteria of not being severely sensory impaired (deaf, dumb or blind), ensured they got medical care first, and then obtained informed consent from them to participate in the study”.

Comment 4: Pg10 lines 115-117: How was consent formally obtained?

Response: Thanks for the comment. This information has been described in the ethical consideration section on page 10 lines 178-182. It is described as: 

“Participants who agreed to participate in this study appended their signature or thumbprint (for those who could not read or write) on the consent form. The consent form translated to the local language (Runyankole) was read aloud to individuals who could not read and write, signed in the presence of a trusted witness (fluent in reading and writing) who countersigned”.

Comment 5: Pg10 lines 119-120: What do you mean 'fast tracked them to receive medical care'?

Response: We have changed the language to make it friendly for readers. Line 127. “Fast track” means ensuring participants get medical care quicker. It has been explained as: “ensured they got medical care first”.

Comment 6: Pg10 line 130: What do you mean by the terms 'lame' and 'crippled'?

Response: In Page 8 Line 147: the terms orthopedically handicapped has been used to make a clearer description of lame and crippled. Thank you. 

Comment 7: Pg11 line 145: What do you exactly mean by this, and what information did they give exactly?

Response: Thanks for the comment. Using the questions adapted from the National Research Council on elder mistreatment monograph, individual types of abuse and the preparator (who abused the elderly) were assessed. However, the information about the preparator was optional (participants who were willing to report, were asked). This has now been rectified in the revised manuscript as well as provided in the questions in Supplementary File 1.

Comment 8: Pg11 lines 146-159: Did you look at higher levels of functionality, such as the ability to complete the instrumental activities of daily living, like managing one’s own finances? If not, why not? This is particularly important when you are considering financial abuse.

Response: The Barthel Index captures the level of functionality on a scale ranging between complete dependent and higher scores - showing more independence (lines 166 to 175-page 9). The tool captures both management of activities of daily living and functionality. However, we do agree with the reviewer that finance is very important in relation to abuse. Unfortunately, we were not able to collect finance-related information as this was not captured by the scale and did not think in such a way, as the reviewer highlighted. We prefer to point out this as our study limitation and hope to consider it in future studies. 

Comment 9: Pg12 line 207: Please define what a VIF is. 

Response: This has been rectified as suggested on page 16 line 226. Thanks. 

Comment 10: Pg16 line 337: But this was not a hospital sample, it was from a sample derived from an outpatient clinic based in a hospital setting.

Response: We have adjusted this to community older adults attending a hospital outpatient department. Thanks for spotting this. 

Comment 11: Pg16 lines 344-351: I am not sure this adequately explains your findings - are there any reasons to explain your contradictory findings?

Response: Thank you for the comment. Educated elderly in Uganda is more likely to be economically well off than the uneducated having accumulated wealth from formal jobs and may also be receiving a pension, therefore, becoming prey to financial abuse by their dependents. This has been explained in the text in lines 375-378.

Comment 12: Pg18 lines 445-449: Additional limitations include the exclusion of people with more severe neurocognitive disorders, whom are at higher risk of abuse

Response: This has been added as suggested on page 30 lines 479-486. Thank you.

Comment 13: Pg18 line 446: See my comment earlier about this being based in the Outpatient setting, not the hospital.

Response: Thank you. We have revised this as suggested throughout the manuscript.

Comment 14: Pg18 Lines 451-455: Please expand more on this section, including a summary of the findings, and future opportunities.

Response: The discussion has been adjusted as suggested. Thanks for your helpful comments. 

Response to Reviewer 2’s Comments

Comment 1: It is stated that about one in ten elderly individuals experience abuse. This estimate is somewhat on the lower side; a recent systematic review of elderly people in rural areas suggests that the figure is closer to 33% (i.e., one in three) – cf. Zhang et al., 2022. This could be reflected in both the abstract and the text, as this study is based on a sample of elderly people from a rural area.

Response: We are thankful to the reviewer for their detailed comment, which has helped to correct one of the most important information we presented wrongly. However, we have now discussed the suggested meta-analysis. Thanks. 

Comment 2: Has this study examined only abuse, neglect, or both abuse and neglect? The title and abstract refer to abuse, but there is a mention of neglect in line 43.

Response: We have added neglect to the study title to reflect the content of the manuscript that covers both abuse and neglect. Thanks for the comment. 

Comment 3: The authors could incorporate recent systematic reviews of elder abuse, particularly from rural or non-Western countries, into this part of the text. See Zhang et al. (mentioned above), Wang et al. (2022) or Ho et al. (2017).

Response: Thank you for this comment. We have now extensively discussed the suggested literature. Kindly refer to the introduction in lines 71-76.

Comment 4: It would be informative to discuss those unique aspects of Ugandan culture that could either contribute to or protect against elder abuse. (For example, if respect for elders is a national cultural value, this could act as a protective factor; on the other hand, if there is a “culture of silence” about violence against women, as in many Asian and African cultures including this reviewer’s, this could be a risk factor for elderly women specifically.)

Response: Thank you for this comment. An aspect of the Ugandan culture has been included in the introduction section’s text on page 4, lines 63-70.

Comment 5: Was the sample size of n = 363 sufficient to address the research questions? 

Response: Yes, the sample size was sufficient as determent based on the Kish and Leslie formula at a precision of 5% using a recent prevalence by Bigala et al in Makifeng county in rural South Africa – a population similar to the current study. [1]

Comment 6: Were any subjects excluded based on criteria other than neurocognitive impairment (e.g., refusal of consent by the individual or by their family members?)

Response: Unfortunately, we did not track potential participants who declined due to other reasons. However, participants who were severely sensory impaired such as the dumb, blind and deaf were also excluded. Thank you.

Comment 7: Terms such as “lame” and “crippled”, though vivid and descriptive, may be considered pejorative or “politically incorrect” by many modern readers. It is preferable if the authors use medical terms (e.g., orthopedically handicapped).

Response: The terms have been substituted as suggested on page 8 line143. Thank you.

Comment 8: Did the instrument used to measure elder abuse require any modifications/adaptations based on local culture or idioms? If so, this could be mentioned. Did any of the study participants find it difficult to understand some of the concepts covered by this instrument?

Response: No, the instrument did not require any modification. The questions were mostly direct and understandable although they were other means of abuse that could have been added that were culturally sensitive therefore a limitation since it has never been used in our context. It has been included among the limitations on page 31 lines 492-493. 

Comment 9: Details of translation and validation of instruments should be provided in the methods section.

Response: Yes, the tools were translated and back-translated to ensure the meaning was maintained. This has been indicated in lines130 to 131. Thanks. 

Comment 10: Were any practical difficulties faced by the researchers in the course of interviewing each subject for 40 minutes (e.g., hearing impairment, fatigue, emotional lability, catastrophic reactions)?

Response: Yes. There were difficulties encountered such as trouble following instructions due to hearing impairment and visual problems. The research assistants dealt with these limitations by assigning more time to these interviews and ensuring maximum sensory stimulation such as speaking louder. We have added them to the methods section and how the research assistants dealt with them on page 7 lines 131-133. Thanks for the comment. 

Comment 11: Apart from measures of functional capacity, did the authors collect any data on the mental health of the participants (e.g., symptoms of depression or anxiety)? This is important because there is a close relationship between elder abuse and mental health.

Response: Thank you very much for the insight. We do agree with the reviewer’s perspective, that is, mental health is closely related to elder abuse. Unfortunately, we did not assess for mental health symptoms. It is something to look at for future studies.

Comment 12: Were the MMSE scores also included in the data analysis? (There is some evidence that cognitive impairment, even if not in the severe range, could be a risk factor for elder abuse – see Lachs et al., 1997)

Response: Thanks for the comment. Neurocognitive results were not included in the analysis. This has been included as a limitation in lines 485-487. 

Comment 13: Was any information collected on family income /socioeconomic status? This could also be a potential risk factor.

Response: A few aspects of socioeconomic data were captured such as the type of housing, employment status, and residence (rural vs. urban). However, more factors could have explained elderly abuse better such as the amount of income earned or the type of savings used. We have included this in the limitation and suggested future studies to add such variables on page 31 lines 486-491. Thanks for the comment. 

Comment 14: If no analysis for neurocognitive results include as limitation

Response: Thanks for the insight. It has been included as a limitation on page 29.

Comment 15: Was information available on the different types/symptoms of physical illness in relation to elder abuse (e.g., were specific illnesses associated with visual or hearing impairment, incontinence, etc. associated with a higher risk of abuse)?

Response: Thank you for the comment. The analysis did not consider specific symptoms. We hope the reviewer understands that this manuscript is already overwhelmed with many analyses and considering specific symptoms cannot be feasible. 

Comment 16: Were any meaningful differences in predictor variables for abuse/neglect identified when comparing elderly men and women (i.e., were there gender differences in risk factors)?

Response: Thank you for the comment. No, there were no gender differences present in this study. The findings resonated with the bivariate analysis. 

Comment 17: This section should be more than a repetition of the study results. For example, the authors could compare their finding of a prevalence of 89% with other studies from rural areas, which yield an estimate of 4 – 61%, and explore the possible reasons (methodological as well as actual) which could account for such a high estimate. 

Response: Reasons possible for the high prevalence estimated in our study are discussed in the text lines 358 - 366. (section elderly abuse). Thanks for the very important suggestion. 

Comment 18: The authors could also discuss recommendations for the prevention and management of elder abuse based on their study results (e.g., based on the predictor variables identified in the multivariate analyses) as well as the limitations/difficulties associated with managing this problem in a rural Ugandan setting.

Response: Thank you for the suggestions. The discussion has been adjusted to include ways of preventing elder abuse, the challenge in prevention, and further recommendations in lines 502-511. Thanks for your helpful comments. 

1. Bigala P, Ayiga N. Prevalence and Predictors of elder abuse in Mafikeng Local Municipality in South Africa. Asian Population Studies. 2014;28:463-74. doi: 10.11564/28-1-500.

---

## [Decision Letter · Decision Letter 1]

10 Jan 2023

Factors associated with elder abuse and neglect in rural Uganda: a cross-sectional study of community older adults attending an outpatient clinic

PONE-D-22-24400R1

Dear Dr. Kaggwa,

We’re pleased to inform you that your manuscript has been judged scientifically suitable for publication and will be formally accepted for publication once it meets all outstanding technical requirements.

Kind regards,

Thalia Fernandez, Ph.D.

Academic Editor

PLOS ONE

Additional Editor Comments (optional):

Reviewers' comments:

Reviewer's Responses to Questions

**Comments to the Author**

1. If the authors have adequately addressed your comments raised in a previous round of review and you feel that this manuscript is now acceptable for publication, you may indicate that here to bypass the “Comments to the Author” section, enter your conflict of interest statement in the “Confidential to Editor” section, and submit your "Accept" recommendation.

Reviewer #1: All comments have been addressed

Reviewer #2: All comments have been addressed

2. Is the manuscript technically sound, and do the data support the conclusions?

Reviewer #1: (No Response)

Reviewer #2: Yes

3. Has the statistical analysis been performed appropriately and rigorously? 

Reviewer #1: (No Response)

Reviewer #2: Yes

4. Have the authors made all data underlying the findings in their manuscript fully available?

Reviewer #1: (No Response)

Reviewer #2: Yes

5. Is the manuscript presented in an intelligible fashion and written in standard English?

Reviewer #1: (No Response)

Reviewer #2: Yes

6. Review Comments to the Author

Reviewer #1: (No Response)

Reviewer #2: The revisions made by the authors are satisfactory. I have no further major changes or corrections to suggest.

7. PLOS authors have the option to publish the peer review history of their article (what does this mean?). If published, this will include your full peer review and any attached files.

Reviewer #1: No

Reviewer #2: **Yes: **Ravi Philip Rajkumar

---

## [Editor Report · Acceptance letter]

16 Jan 2023

PONE-D-22-24400R1 

Factors associated with elder abuse and neglect in rural Uganda: a cross-sectional study of community older adults attending an outpatient clinic 

Dear Dr. Kaggwa:

I'm pleased to inform you that your manuscript has been deemed suitable for publication in PLOS ONE. Congratulations! Your manuscript is now with our production department. 

Kind regards, 

on behalf of

Dr. Thalia Fernandez 

Academic Editor

PLOS ONE